# Studies on Hydrological Processes on Karst Slopes for Control of Soil and Water Loss

**Zehui Wang [1,2,†], Ding Luo [1,2,†], Kangning Xiong [1,2,*], Xing Gu [1,2] and Zhenzhen Zhu [1,2]**

[1] School of Karst Science, Guizhou Normal University, Guiyang 550001, China; wangmuyang1130@163.com (Z.W.); weinan20081208@163.com (D.L.); gggggxing@163.com (X.G.); zhuzhenz0910@163.com (Z.Z.)

[2] State Engineering Technology Institute for Karst Desertification Control, Guiyang 550001, China

[*] Correspondence: xiongkn@163.com

[†] These authors contributed equally to this work.

**Abstract:** Soil and water loss in karst areas seriously restricts the sustainable development of karst ecosystems, the economy and society in southwest China, which has been a concern of, and studied by, many scholars in China and abroad. Soil and water loss has a great influence on the evolution of rocky desertification, groundwater quality, drought and flood disasters in karst regions. This paper aimed to provide a review of studies of slope hydrological processes in soil and water loss. In this paper, 322 related articles retrieved from the Web of Science database and CNKI database were systematically reviewed. Firstly, a quantitative study was conducted to analyze the annual number, countries and research progress of the published literature. Secondly, the main progress and achievements of slope hydrology and soil erosion control technology were classified and summarized according to theoretical research, mechanism research, technology research and technical demonstration. Finally, the key problems and future research prospects are put forward, starting with the existing technology, in order to find more suitable soil and water loss control measures in karst regions and achieve economic and ecological benefits.

**Keywords:** karst; slope hydrology; slope hydrological process; water and soil loss





## 1. Introduction

Karst covers an area of 22 million km$^2$, which accounts for 15% of the earth's land surface in total. It is home to about 1 billion people [1,2], mainly distributed in southern China, in which Guizhou Province serves as the center, south-central Europe and eastern North America. The carbonate outcropping area in the karst region of southern China is the largest in the top three continuous regions of the world (Figure 1) [3–5].

The hydrological process of sloping land mainly includes precipitation, infiltration and runoff [6]. Precipitation falls to the surface, forming runoff on the surface. During this process, water penetrates in different forms, forming a whole hydrological process on the slope. Soil and water erosion refers to the destruction and erosion of soil, water resources and land productivity under external forces such as water power, gravity and wind, including surface erosion and soil and water loss [7]. Soil and water loss is a global problem. Scientists in various countries have carried out extensive research, and many important achievements have been made in terms of the process and mechanism of soil and water loss, model building, sensitivity evaluation and treatment measures since the beginning of the 20th century [8,9].

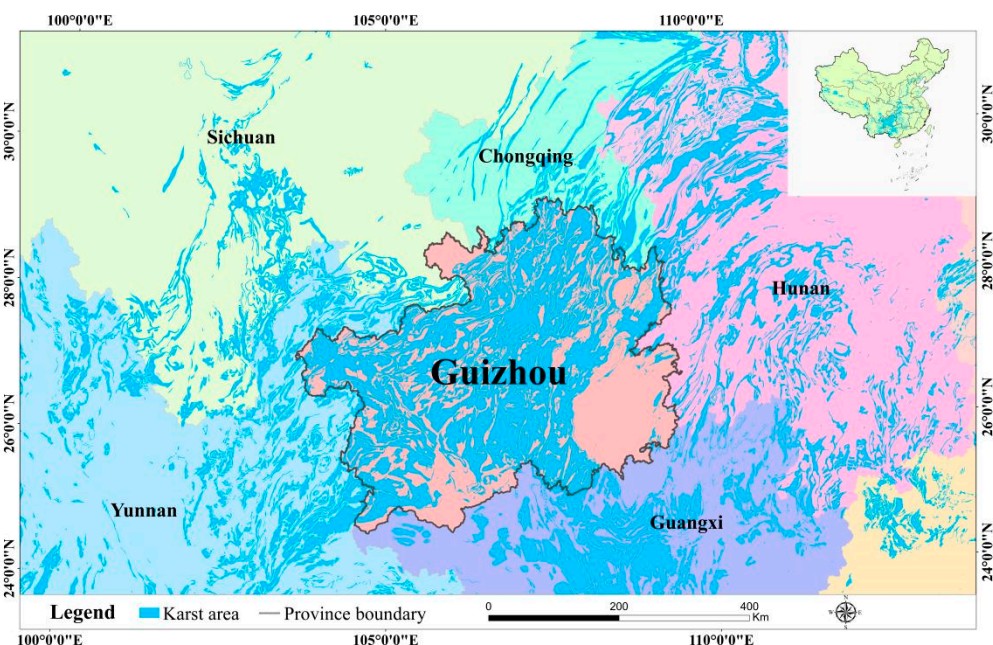

**Figure 1.** Partial layout of karst in South China centered on Guizhou Province.

In the geographical phenomenon of soil and water loss, sloping land is the most basic geographical unit, because human activities are more frequent in these areas, but the slow speed of soil formation in karst regions and the shallow soil layer, combined with the impact of heavy rain and human activities, make the contradiction between humans and land more prominent, and the serious reclamation of sloping land aggravates the land and water loss [10,11]. Restricted by geological conditions, karst regions are short of water and soil resources [12]. The particularity, fragility and complexity of karst ecological-geological environments determine the complexity and diversity of soil and water loss processes in karst regions. Soil and water loss in karst regions is restricted by many factors and has certain characteristics. The most important causes of soil and water loss are cultivation without soil and water conservation and the development of karst fissures, water holes and underground rivers, which are affected by tectonic landforms, lithology, climate, soil properties, vegetation and underground systems [13–16]. Therefore, the study of hydrological processes on sloping land is the theoretical basis of soil erosion control. Many achievements have been made in the process and mechanism of soil erosion and prediction and erosion models in the karst regions of southern China. However, compared with the more mature soil erosion control studies in the Chinese Loess Plateau (CLP) and other regions, research on the combination of slope hydrological processes and soil erosion is still in the development stage [17].

The development and application of soil and water erosion control technology are of great practical significance to the development of basic agricultural economies, the restoration of ecological environments and the sustainable development of karst regions.

## 2. Materials and Methods

This paper conducted a literature retrieval based on the China National Knowledge Infrastructure (CNKI) database and Web of Science (WOS) core database. Firstly, "slope hydrology" or phrases including "slope hydrology" were entered into the search item "Subject" for primary retrieval in the CNKI database, and "soil and water loss" or "soil and water loss" was entered into the search item "full text" for secondary retrieval. The retrieval deadline was December 2021. The retrieval time range was the maximum retrieval time range of CNKI. The scope of the literature was all journal databases. Then, "Slope Hydrology" was entered into the retrieval item "Topic" in the Web of Science core database for initial retrieval, and "Soil and water Loss" was used for secondary retrieval in this

result. The search deadline was December 2021. Finally, according to the research purpose of identifying "characteristics of relationship between slope 'hydrological process' and 'soil and water loss'", the Chinese and English studies retrieved were manually screened. Through the above retrieval and screening, a total of 322 articles were obtained: 21 Chinese journal articles, 1 conference article (1 domestic conference, 0 international conferences), 24 master's theses and 30 doctoral theses, 7 patent inventions and 239 research papers published in international journals.

Based on these results, we analyzed the literature using statistical analysis software. The literature statistics were conducted in Excel, the graphs were made using Origin software and the analysis performed in both allowed us to obtain the distribution of the literature by year, content and region and to identify different trends in the literature growth.

### 2.1. Document Analysis

According to the recordings and statistics, research on underground soil and water erosion in karst regions was first found in the 1960s. Liu Zhigang first proposed the concept of "underground soil and water erosion" [18] in 1963 when he studied soil erosion in Du'an County, Guangxi Province. Before the year 2000, there were few studies on karst soil and water loss. Scholars and experts paid more attention to the occurrence and development of surface soil and water erosion, rock desertification and restoration of fragile ecological areas. According to the research and Figures 2 and 3, research on the hydrological processes and soil and water loss on sloping land can be divided into three stages. The period of 1960–2000 is the first stage, during which research on soil and water loss was in the phase of germination, mainly focusing on the mechanism of soil and water loss. The period of 2000–2010 is the second stage, during which research on soil and water loss entered the emerging stage. At this stage, scholars began to seek reasonable technical methods to prevent and control soil and water loss. In the third stage, from 2010, attention has been paid to soil and water loss in China and abroad, and there are more special studies abroad which study the mechanism of soil and water loss through simulation experiments. (Figures 2 and 3).

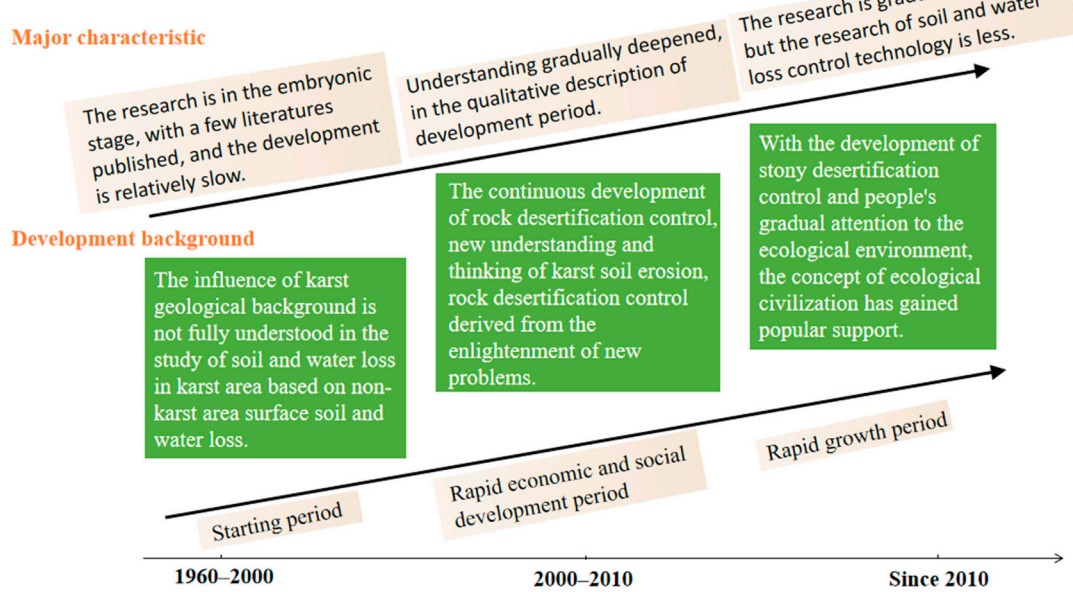

**Figure 2.** Stage division of research on hydrological processes and soil erosion on sloping land.

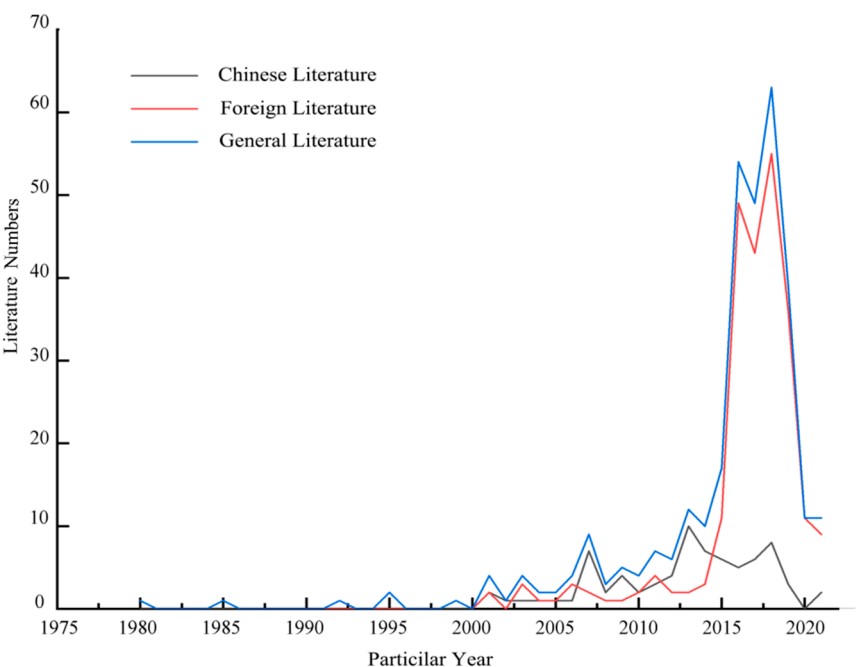

**Figure 3.** Distribution of the domestic and foreign research literature.

### 2.2. Regional and National Distribution of the Literature

Among the 239 research papers published in international journals, studies on slope hydrology and soil and water loss include tropical, temperate and cold zones, with more studies in developed countries and less in developing countries (Figure 4). Studies on "slope hydrology" and "soil and water loss" in the United States and China are the largest, accounting for 66.95%, followed by Spain and New Zealand. There are some studies in Ethiopia, Iran, Italy, Canada, Brazil, France, Japan, Tanzania, Russia and Switzerland. In addition, there are also a few studies in Mexico, India, Korea and other countries (Figure 4). The top six authors in number of literature published on the topic are Pan Chengzhong (5), Zhang Fengbao (3), Fu Bojie (3), Fang Nufang (3), Miller, Marcus E. (3) and Yu, Bofu (3).

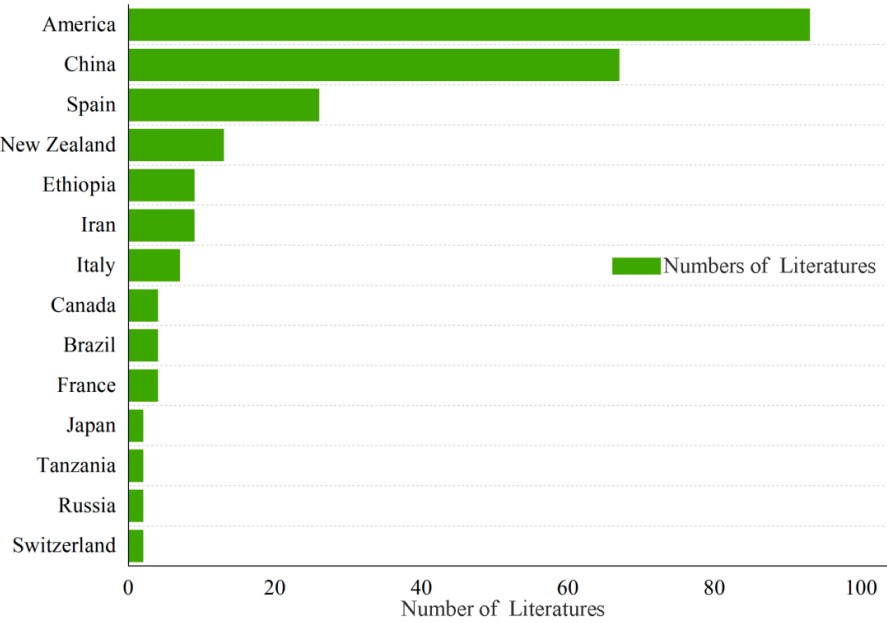

**Figure 4.** World literature research area: hydrological processes and soil erosion on sloping land.

*2.3. Distribution of Research Type*

All research types in the literature were classified and summarized as theoretical research, mechanism research, technology research and technical demonstration. Among them, the theoretical research literature accounted for 54.66%, the mechanism research literature accounted for 22.05%, the technology research literature accounted for 19.57% and the technical demonstration literature accounted for 3.73%, where the theoretical research and mechanism research literature accounted for the mainstream. In the early stage of the research on "slope hydrology" and "soil and water loss", related theory and review papers dominated, which focused on the concept and theory of slope hydrology and soil and water loss. However, most of these studies were only simple conceptual and theoretical research, and mechanism research on soil and water loss had not been conducted. With the deepening of the research, the literature on technical methods began to increase; in recent years, application demonstration research has begun to rise, focusing on a variety of soil and water loss control technologies for demonstration. The change of research content from theoretical research to technical methods and then to application demonstration enriched the study of slope hydrology and soil and water loss (Figure 5).

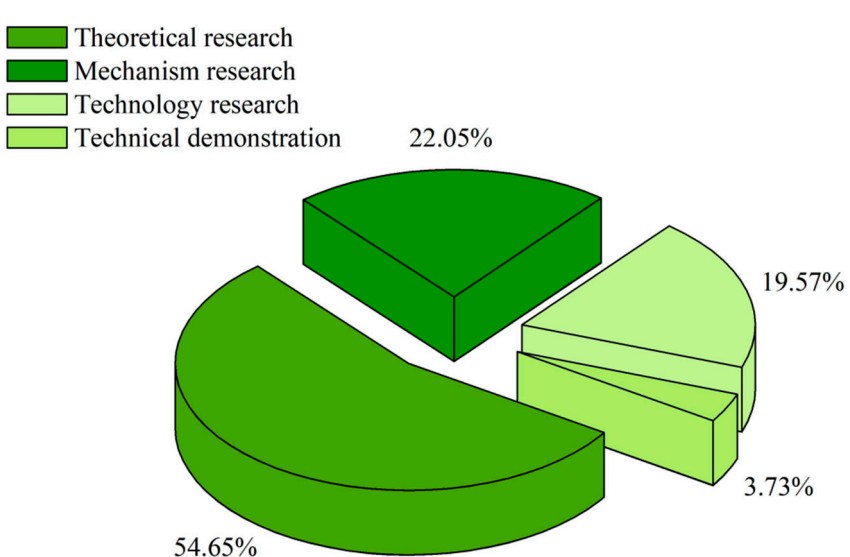

**Figure 5.** Classification of document types on sloping land hydrology and soil erosion.

## 3. Major Progress and Landmark Achievements

*3.1. Theoretical Research*

3.1.1. Rainfall and Infiltration

Most studies on water balance focus on soil infiltration. Soil infiltration refers to the process of water infiltration from the soil surface into the soil to form soil water, mainly including the infiltration and seepage stages. Understanding the rainfall infiltration performance of soils is of great significance to the study of flood processes, soil erosion, soil moisture and the layout of soil and water conservation prevention measures [19]. When the rain intensity is less than the infiltration capacity, all rainfall is absorbed by the soil. When the rain intensity is greater than the infiltration capacity, the absorption rate can only equal the infiltration capacity, and the rest is runoff generation [20]. Under different rainfall intensities, the form of the infiltration curve is the same. If the rainfall duration is long enough, the stable infiltration rate and total infiltration of homogeneous soils have nothing to do with the rainfall intensity, but the instantaneous infiltration rate is greatly affected by the rainfall intensity and the temporal change in the rainfall intensity [21].

3.1.2. Soil and Water Loss on a Slope Is a Very Complex Process

At present, it is widely believed by scholars in China and other countries that sediment generation on sloping land is a very complex process, which is controlled by many fac-

tors, such as field rainfall characteristics, underlying surface conditions, soil physical and chemical properties, slope length, soil erosion resistance and tillage systems [16]. Splash and sheet erosion are the main drivers of slope soil erosion and rocky desertification. Their process includes Soil particle separation in topsoil, sediment transport and sediment concentration processes. The most direct consequences of water loss include barren soil, soil fertility decline, loss of land productive potential and even permanent loss of production potential, which would lead to rocky desertification landscapes of different degrees [22]. In 1965, Wischmeier and Smith established the well-known universal soil loss equation ULSE (Universal Soil Loss Equation) [23]. This equation is mainly used to study the prediction of soil erosion in cultivated areas, without taking the process and mechanism of soil erosion into consideration. In addition, it was mainly adapted to cultivated areas on the plain, but many revised versions were later proposed [24]. After the 1980s, Britain, Belgium and Holland successively developed the European soil erosion model, which is suitable for large plain areas but not suitable for karst regions in China, especially in southwest China where the arable land area is relatively broken. With the development of computer technology and geographic information system technology, international scholars have used RS (remote sensing) and GIS (geographic information system) technology to study the spatial–temporal dynamic changes of regional soil and water loss [25]. This method has the advantages of large spatial and time scales, and it has good timeliness for monitoring geographical national conditions and ecological water and soil control. However, soil and water loss in karst regions has a large proportion of leakage from the underground space through pipelines and hidden underground rivers. In karst regions, RS and GIS technology can be used to dynamically monitor soil erosion and analyze land use types, which is of great scientific significance to soil erosion monitoring in karst regions [26,27].

### 3.1.3. Unique Forms of Soil and Water Loss in Karst Regions

Different scholars have different definitions of unique forms of soil erosion in karst regions in terms of language expression, but the general meaning is the same. Soil and water loss is a phenomenon where the soil physical structure changes and migrates to karst pipelines, fissures and underground rivers under the action of water power and gravity, resulting in soil and water loss [28]. Karst soil and water loss is a process in which surface debris and soil migrate or transport to the underground system by means of downpour, creep and collapse along the fissure channels developed in the bedrock under the action of surface water erosion and gravity [29]. Karst soil and water loss is a process where the soil parent rock and surface soil enter the karst underground aquifer along the karst solution groove, solution groove, depression and rock fissure. In this process, the soil which covers the underground cracks (holes), and pipelines in karst areas are eroded by gravity, such as creep and staggered erosion [30,31].

### 3.2. Mechanism Research
### 3.2.1. Important Factors Affecting Soil and Water Loss in Karst Regions

Soil, as the basic, precious natural resource in karst areas, has a shallow soil layer, a discontinuous soil cover and a sporadic distribution with a thickness of about 30 cm, which is the direct object of soil and water loss. The soil in karst regions has a low water storage capacity, and it is easy to dry and disintegrate when exposed to water, thus forming runoff migration with atmospheric precipitation. Moreover, C layers are usually absent in soil profiles in karst regions. There are different soft and hard interfaces between the carbonate parent rock and soil. Thus, there is no tight contact surface between the rock and soil. Under the condition of rainfall, soil particles will leak into caves and underground streams through fissures, sinkholes and/or other tunnels developed in carbonate rocks. It can be seen that soil properties and the karst rock–soil interface are important factors affecting soil and water loss. Vegetation is the most sensitive factor to soil erosion and plays an important role in preventing and controlling soil erosion. The coverage rate and types of vegetation affect the occurrence and development of karst soil and water loss.

After the destruction or disappearance of surface vegetation, the original fixation ability of roots to the soil weakens or disappears, aggravating underground water and soil loss [32]. Vegetation can not only improve the soil structure and enhance the soil water infiltration capacity but can also improve soil erosion resistance, so there would be a lower occurrence of soil and water loss [33].

### 3.2.2. Characteristics of Rainfall and Sediment Yield

Rainfall is an important factor affecting the process of soil and water loss, and it is the main driving force of rainfall splash erosion, slope runoff and underground loss [34]. Rainwater can act as a solvent of soluble nutrients. When the rainfall exceeds the runoff yield threshold and forms slope runoff, rainwater is also a medium carrying other forms of nutrients. In this process, rainwater can act as a lubricant and promote the transfer of the soil to underground fissures and karst pipes through peristalsis. Therefore, rainfall characteristics (duration, intensity, amount, etc.) are important factors affecting sediment yield and ion loss on karst slopes, and many scholars have proved that the rainfall intensity significantly affects sediment yield on slopes [35]. For example, Mohamadi used the method of runoff plots to observe the influence of field rainfall on runoff and sediment production and concluded that the relationship between soil loss and rainfall intensity was linear in the case of a low rainfall intensity, while the function tended to be non-linear in the case of a high rainfall intensity. In addition, the higher the rainfall intensity, the more intensive the erosion and the higher the slope runoff [36]. Fu et al., after studying the runoff generation event of a dolomite karst slope in a subthermal belt based on a rainfall simulator, found that the response of the slope and underground runoff generation was more sensitive under a high rainfall intensity than under a low rainfall intensity, which was embodied in a shorter runoff generation time. When the rainfall condition reached a certain runoff generation threshold, slope sediment and subsurface flow did not start [37]. By directly affecting slope rainfall runoff, and subsequently slope sediment yield and total ion erosion, the amount of rainfall is lower than the osmotic quantity at the beginning. The wetting of the surface soil mainly causes it to become saturated. Thus, there is no slope runoff. When the rainfall continues, the soil moisture is saturated. Then, soil particles exfoliated by raindrop spillage block soil pores, so that the subsurface leakage decreases and slope runoff begins to increase, which eventually lead to slope erosion and an increased nutrient carrying capacity.

### 3.3. Technology Research

### 3.3.1. Tracer Technology

$^{137}$Cs was first introduced to China from abroad by Zhang et al. in the 1980s, who used this technology to quantitatively study soil erosion in the Chinese Loess Plateau region [38]. $^{137}$Cs is produced by atmospheric nuclear explosion and falls to the surface under the action of rainfall and its own settlement. It is attached to surface soil organic matter and clay particles, which are difficult to be dissolved by rain and redistributed with the migration of soil particles [39]. $^{137}$Cs is a good tracer of surface loss and underground loss in karst regions. Wei et al. studied the underground loss mechanism of soil in a karst trough valley in Chongqing using $^{137}$Cs and the matching method. They found that the ratio of surface loss and underground loss was 77.55% and 25.45%, respectively [40]. Bai et al. studied the process of soil erosion in the karst area of central Guizhou using the $^7$Be tracer based on the characteristics of continuous subsidence and the short half-life of the isotope $^7$Be, which can trace the migration of short-term and seasonal particles [41].

### 3.3.2. Sloping Terracing Technology

Sloping terracing is one of the main ways to realize sustainable agricultural development in karst areas. It can improve economic benefits, resource efficiency, environmental benefits and social benefits at the same time. It is widely welcomed by farmers because it can significantly increase the arable land and the yield per unit area. It is a comprehensive

project which integrates water conservancy, agriculture, forestry, finance and poverty alleviation [42]. Terrace technology mainly includes terrain adjustment, land leveling, soil layer thickening, the construction of "three ditches" (water diversion ditch, drainage ditch, along the valley), the "three pools" (impounding reservoirs, grit tanks, septic-tank) such as in slope drainage, the construction of small projects for soil and water conservation, the development of rainwater harvesting facilities, to build"flat, thick, soil, solid, fat" level terrace land. According to the overall planning of different ground slopes, climate conditions, natural platform positions, soil-forming parent materials, soil types, soil obstacle factors, planting modes, etc., the boundary between plots is broken and the platform position is set according to the contour line to adjust the terrain. The cost for the construction of ridges is relatively low as it adapts to local conditions and local materials, but it can receive a great benefit after as the transformation can achieve the flat, solid, soil, thick and fertilizer quality standards. Sloping ladders are one of the most widely used engineering and technical measures for soil and water conservation. Due to the shortage of cultivated land, poor land quality, frequent droughts, poor water utilization capacity, serious geological disasters and shortage of funds in agricultural production in karst regions, terracing in karst areas involves the sustainable utilization of natural agricultural resources development, making it one of the most promising technologies. It can take advantage of the development of agricultural land and the increased cultivated land, improve the soil water conservation function and ease the problem of drought and floods. Slope surface soil and water conservation can also reduce the amount of leakage from the underground (Figure 6).

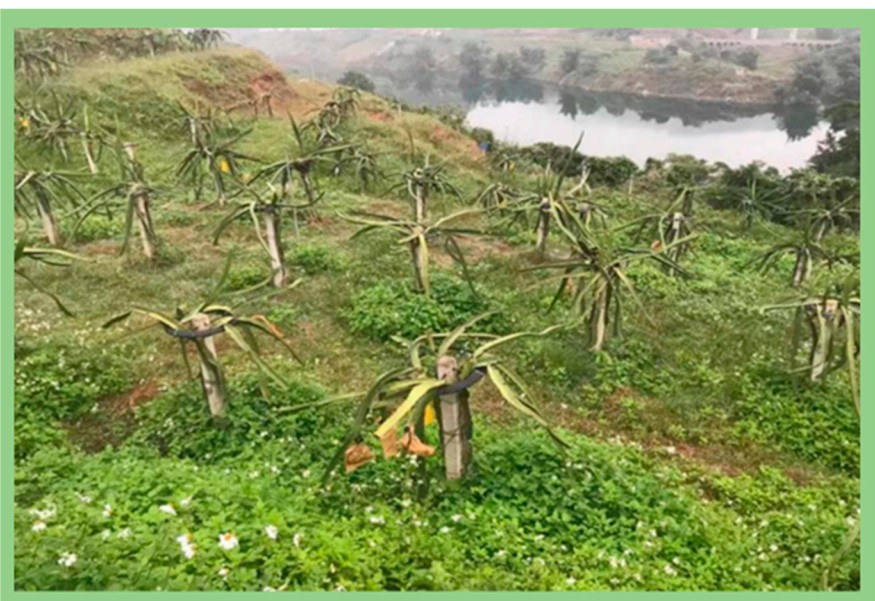

**Figure 6.** Guanling-Zhenfeng Huajiang demonstration zone, Guizhou Province, China.

### 3.3.3. Bioengineering Technology

The biological resistance control measures of karst areas mainly include returning farmland to forests and grassland, afforestation, artificial vegetation restoration and soil and water conservation measures by using technologies such as plant-expanding green manure, bare plant hedges and sloped partition cropping. These measures are based on the principle of the rainfall redistribution effect on the vegetation, which would weaken the raindrop kinetic energy, and the effect on soil improvement in particular. This effect manifests in enhancing soil permeability, water holding capacity and anti-erosion and anti-dispersion ability, thus reducing the occurrence of surface soil erosion.

In order to protect and sustainably utilize the land resources in the peak cluster depressions where farmland is concentrated in karst regions, corresponding measures have been adopted to reduce soil and water loss on the surface and in the underground, and

sediment filtration has been undertaken. On the basis of existing work, the principles of effective protection, full and reasonable utilization of water and soil resources and ecological environment improvement have been followed. Different measures have been taken according to different land types, combined with the comprehensive prevention and control of rocky desertification. Biological engineering technology has been adapted in view of the unique hydrogeological structure, rock + soil combination characteristics of different geomorphic parts and the main routes of soil and water loss.

Through the protection of water and soil, we can protect source forests, and the soil as well as the ecological functions of water resources can be improved. According to local conditions, the measures of water source forests, hedgerows, ditch interception, slope terracing change, stone blasting, building walls to protect soil, soil preparation, etc., are implemented with the biological measures as the core method, combined with the corresponding and scattered engineering measures. Attempts are made to take both the ecological and economic benefits into consideration by means of covering the exposed rocks with vines or removing exposed rocks to prevent and control soil erosion caused by rock surface runoff, and improving the planting structure to avoid turning over soil and to reduce the underground leakage of stone corners and thin soil layers. For land with a thick soil layer and a continuous soil cover, the combined methods of stone blasting, building walls to protect the soil, hedgerows and three-dimensional planting are implemented to prevent soil erosion. Seven types of suitable water conservation plants in karst areas were selected, named bobcat bean, pitaya fruit, herbage, honeysuckle, climbing FIG, red cangeteng and Rhizoma pumila. The optimal utilization patterns of the ecological land were designed, such as grass + honeysuckle, grass + dragon fruit, grass hedgerow, cat bean + peanut, fufang rattan + fruit and mountain hemp stick + fruit. Through experiment and demonstration projects, a soil and water conservation technology system with soil loss and resistance control as the core was developed, including the construction technology of soil and water conservation forests, climbing FIG hedgerows, exposed rock sprout hedgerows, ridge hedgerows with a wall and soil preservation, slope-type hedgerows, soil + rock microgeomorphic unit soil and water conservation technology, slope plant ladder technology and pitaya dragon fruit planting technology in depressions [43].

*3.4. Technical Demonstration*

3.4.1. Rocky Desertification Vertical Zoning Model Demonstration Where Rock Desertification Control and Ecological Environment Were Significantly Improved

Based on the research results of the slope, depression sink hole and fissure soil erosion, Zou et al. conducted research on the development of farmland terraces, hedgerows, no-till planting soil conservation tillage technology and karst peak cluster depression control technology of a soil erosion resistance system [44]. They chose rocky desertification areas on a karst slope in the Huanjiang County south pond rural demonstration area in Guangxi Province. The integrated management mode of vertical desertification in the peak cluster depression was implemented according to local conditions and the vertical differentiation law of the slope top, upper slope (stony slope), slope waist (soil stony slope), piedmonts (soil slope) and easily waterlogged depression. The stone slope at the top and top of the slope were closed to restore vegetation and prevent soil loss underground and on the surface. In order to prevent erosion by ploughing, an economic forest and an ecological forest were planted on the slope and rocky slope by returning farmland to forests, irrigation and grassland, aiming at soil slope terracing control and soil erosion. Reasonable water conservancy projects should be built in the lowland to prevent waterlogging and ensure high and stable yields of basic farmland crops. The model has been widely applied in the karst peak depression areas of northwest Guangxi.

3.4.2. The Demonstration Area Should Adapt Measures to Local Conditions

Yin et al. took the Guohua, Nongla and Huanjiang demonstration areas as research objects, which have achieved remarkable results in soil and water conservation and eco-

logical restoration in karst rocky mountainous areas of Guangxi, and summarized the commonness, characteristics and benefits of the three measures of soil and water conservation in each demonstration area [45]. In the fruit demonstration area, they changed the past large-scale planting of corn, soybean and other crops in a single mode and promoted the characteristics of the demonstration area with the drought-tolerant fruit pitaya in the dry land. In order to improve land fertility, the soil was improved by returning straw to the field, applying organic fertilizer from farmers and filtering mud from sugar plants with the sulfite method, and agricultural measures such as plastic film mulching and balanced fertilization were carried out. Water-saving techniques such as sprinkler irrigation and drip irrigation were popularized.

In the Nongla demonstration area, they changed the traditional hillside reclamation mode and chose to focus on developing forest fruits at the foothills and gentle slopes, interplanting medicinal materials and interplanting vines on the steep slope of Shishan. In the Huanjiang demonstration area, they developed the mode of forest agriculture ecological planting, giving full play to local advantages and making full use of the tree and fruit tree space by intercropping various crops in different growth seasons, among which mulberry planting sericulture and cultivation of edible fungi have become the characteristic industry of the demonstration area.

## 4. Key Problems to Be Solved and Suggestions

### 4.1. Theoretical Research

The slope affects the relationship between slope erosion and sediment yield and water balance; the rainfall intensity is the main driving force; and surface runoff is the carrier of soil and water loss and soil nutrient loss. All three are important factors affecting surface erosion and sediment yield, as well as important links of the water cycle and the key to influencing the slope water balance. Understanding the mechanism of slope hydrological processes such as rainfall, sediment yield, runoff yield and infiltration is of great significance for improving and controlling soil erosion. However, the current research on these environmental problems is mainly focused on the treatment of the Loess Plateau in China, and the overall study of slope rainfall, runoff, infiltration and other hydrological processes in karst regions has not been conducted. The hydrological processes of sloping land can be linked and systematically studied.

### 4.2. Mechanism Research

Water is an important environmental medium in the process of soil and water loss, and the hydrological process affects the law of soil and water migration to a great extent. Moreover, as a soil solution, the migration process and physical and chemical properties of the water are also closely related to soil properties. Soil is destroyed, separated, transported and deposited under the hydraulic action of rainfall, runoff and other factors. Meanwhile, the soil itself has an interaction between erosion resistance and water erosivity, and the interaction mechanism of the two is obviously influenced by other natural factors such as rainfall and vegetation. At present, the interaction mechanism of slope hydrology and soil properties, rainfall, vegetation and other factors is studied deeply in karst regions, but cross studies of soil erosion and hydrological processes in karst regions are rarely reported. Therefore, the relationship between slope hydrology processes and soil erosion should be analyzed based on soil properties, vegetation and rainfall characteristics.

### 4.3. Technology Research

There are relatively few studies on evapotranspiration and its slope components and slope position differences in karst regions. Evapotranspiration is an important output term of the water balance. Studies at the slope scale can connect the same vegetation type or similar types with the watershed level, but this is still a weak point at present. The WEPP slope erosion model can simulate soil erosion on a slope, but it cannot complete the simulation

calculation of hydrological processes on the slope. Therefore, it will be easier to calculate evapotranspiration on the slope by constructing a perfect slope hydrological model.

The relationship between the hydrological process and soil and water loss has been established.

### 4.4. Technical Demonstration

Soil and water loss, as a unique form of soil and water loss in karst regions, has an important impact on the evolution of rocky desertification, the quality of underground river water, drought and flood disasters. At present, research on water and soil loss and resistance control in karst areas is mainly focused on theoretical research and simulation tests. On the basis of theoretical research, the control effect of soil and water loss and the resistance of mixed agriculture and forestry were studied, in order to obtain the best combination of mixed agriculture and forestry. On this basis, the development of mixed agriculture and forestry on sloping land can not only maintain ecological benefits but also improve economic benefits. Initial results have been achieved (Figure 7).

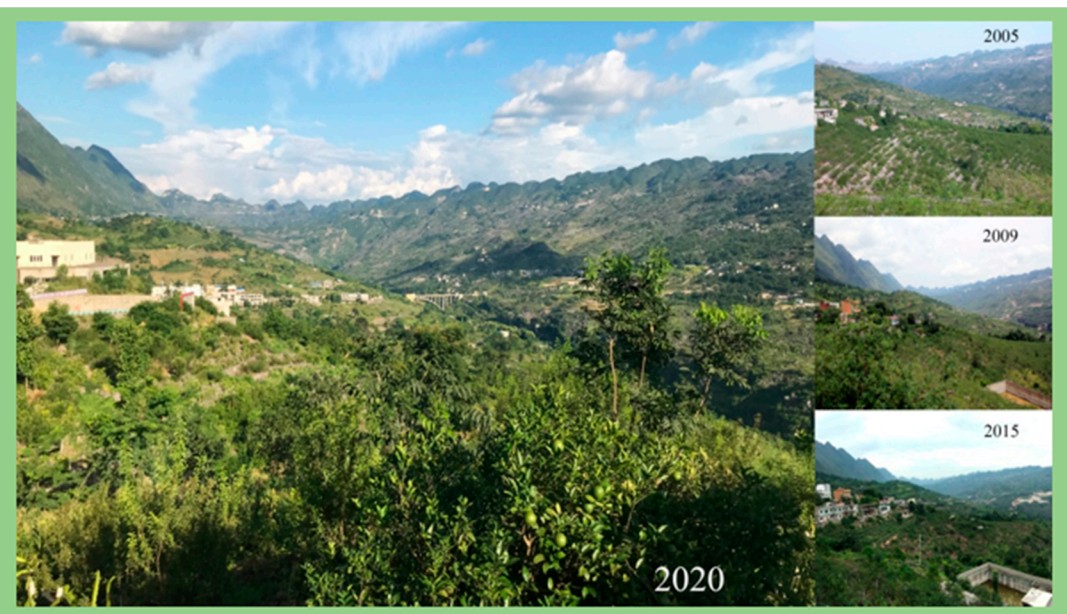

**Figure 7.** Effect of Guanling-Zhenfeng Huajiang research area in Guizhou Province, China.

### 5. Expectation

It is an urgent need for national ecological environment construction to carry out the overall planning and design of sloping land environments on the basis of the research on the mechanism of soil and water loss under the dual structure of karst, which has important theoretical and practical significance to control the development of regional rocky desertification and promote regional sustainable development.

As shown in Figure 8, mixed forestry planning has changed from the traditional strip planting layout perpendicular to the contour line to concentric ring planting parallel to the contour line. On this basis, sloping land is divided into three heights, A, B and C (hat on the top, belt on the mountainside and bedding at the bottom).

Due to the particularity of karst landforms, soil and water leakage and nutrient migration on slopes lead to the imbalance of soil and water nutrients in the three positions at the bottom of the slope and the top of the slope, thus leading to a phenomenon currently existing: "No hat on the top of the mountain, poor belt on the mountainside, only the bottom of the mountain". The focus of our study now lies in:

(1) Secondary forest protection or artificial planting of a secondary forest on the top of slope A, in order to control soil and water loss and resistance and fix soil and water nutrients, so that the "top of the mountain wearing A hat";

(2) Establishing a resistance control method for the rational allocation of mixed forestry on slope B;

(3) Conducting research on the resistance control of the sinkhole at the bottom of slope C, in order to combine economic benefits with ecological benefits.

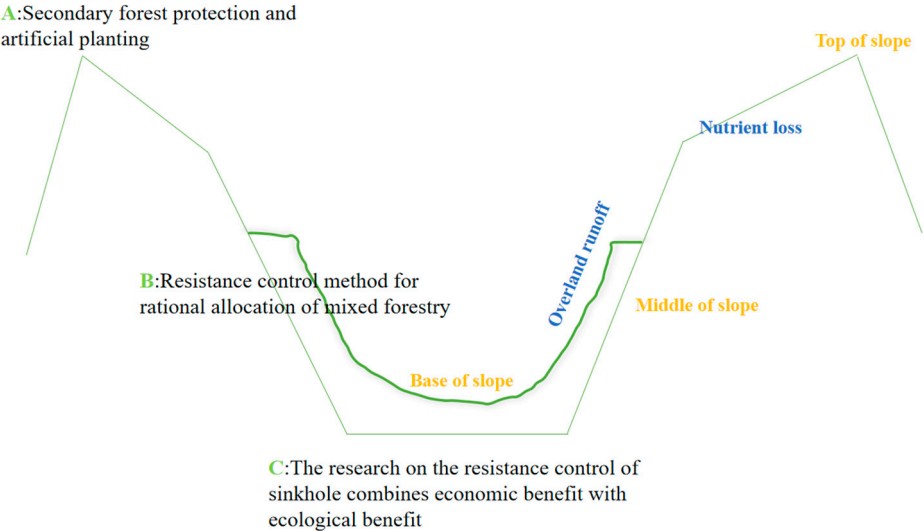

**Figure 8.** Schematic diagram of soil and water loss and resistance control technology section on sloping land.

## 6. Conclusions

At present, soil and water loss and resistance control technologies on sloping land in the karst area of China are still in the exploratory stage, and many of them are based on the previous research results of soil and water loss. For example, water conservation forests have been developed at the tops of mountains; lianas have been planted on steep slopes to make full use of the soil resources in the stone crevices; and shrubbery has been planted on gentle slopes. How to control rocky desertification in karst areas with more targeted technology is a problem worth discussing. The existing soil and water conservation measures cannot completely solve the problem of soil and water loss, and some will even promote soil and water loss and soil stones on karst slopes. The vertical zonation of soil on sloping land requires different measures of soil and water conservation according to local conditions. Therefore, it is suggested that, in future research, the configuration and design scheme of comprehensive control measures of soil and water loss should be deeply studied to solve the problem of soil and water loss on karst slopes.

**Author Contributions:** Conceptualization, K.X. and Z.W.; methodology, Z.W.; software, Z.W.; formal analysis, Z.W. and D.L.; investigation, X.G.; data curation, Z.W. and Z.Z.; writing—original draft preparation, Z.W.; writing—review and editing, Z.W. and D.L.; project administration, K.X.; funding acquisition, K.X. All authors have read and agreed to the published version of the manuscript.

**Funding:** This study was supported by the Key Project of Science and Technology Program of Guizhou Province (No. 5411 2017 Qiankehe Pingtai Rencai), the World Top Discipline Program of Guizhou Province (No. 125 2019 Qianjiao Keyan Fa) and the China Overseas Expertise Introduction Program for Discipline Innovation (No. D17016).

**Institutional Review Board Statement:** Not applicable.

**Informed Consent Statement:** Not applicable.

**Data Availability Statement:** Data are contained within the article.

**Conflicts of Interest:** The authors declare no conflict of interest.

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
