# Peer review of "Studies on Hydrological Processes on Karst Slopes for Control of Soil and Water Loss"

_sustainability, doi:10.3390/su14105789_

Round 1

Author Response

Dear reviewer,

I have re-thought and revised the article in depth according to your questions and suggestions, and I have given detailed replies to each question. Thank you for your suggestions on the article. I have learned a lot. Please refer to the attachment for specific answers.

Reviewer 2 Report

Title:

the title should be rephrased to better describe the present review study

Introduction session:

The second paragraph of the introduction “Slope land is the basic geographical………..

 tillage measures and integrated management modes” : the scientific evidences in karst context and under others contexts are not clear.

The third paragraph “The hydrological process of sloping ……………. the beginning of the 20th century [11-12] “no interest to present a basic informations for erosion process on the introduction session, especially for a review paper this paragraph must be rewritten, what is the specificity for karst region, it is relevant to precise this specificity, are there some scientific results under this context and are they different compared to other context?  

The fourth paragraph “The ecological environment in karst areas …………………… into soil and water loss [13]” must be amended by some references in order to highlight the importance of this study in the karst conditions.

Materials and Methods session

Why for the information retrieval, authors did not use “karst” for their research of articles.  

Figure 3: x axis, is it the number of the research papers or the sites, some authors studied more than one site, us the cited review papers 15, 16 and 35?

What does mean “theoretical research”?

What is the difference between “theoretical research” and “mechanism research”?

Some clarification about the adopted methodology of stratification of the used papers must be added (spatial level, approach, ...)

Figure 4: is not the appropriate type of the graph to present these results

Major progress and landmark achievements:

Titles : 3.1.1, 3.1.2 and 3.1.3 must be simplified as title format or deleted 

3.1.1: very basic information with no add value for the present study, nowadays, using modelling approach to study soil erosion is not new way.

3.1.2: basic informations linked to the parameters used for RUSLE equation  

What does mean “RS” abbreviation must be identified previously.

The references linked to the study of the erosion in sole karst regions are limited

3.2. the magnitudes of some scientific results must be added

3.3.1. what it the difference of these method, cost, applicability, spatial unit, precision, ?

3.3.3, references should be added to this section.

3.4. there some confusions between technical observation or recommendations and obtained scientific results.  

Key problems to be solved and suggestions

Titles : 4.1.1, 4.1.2, …. must be simplified as title format or deleted 

This section describes some technical recommendations for very local conditions, these recommendations should be justified by research results.

some comments are added on the pdf format    

Author Response

(The authors gave the same response as above.)

Reviewer 3 Report

The MS “Implications of advances in hydrological processes on karst slopes for control of soil and water loss and resist” is addressing a long debated problem of slopes. The MS is interesting for regional readers in specific and global readers in general. My comments are as under:

  1. Title is quite long, it may be shortened
  2. Abstract may highlight some of the study output instead of just writing quantity of reviews and so on.
  3. Mention slope % age in the abstract. Please restructure the abstract indication problem statement, reasons of problem and some solution. Currently abstract have some useless statement such as “The slope is where human activity is concentrated”.
  4. Introduction have statement “ Slope land is the basic geographical unit when it comes to the geographical phenomenon of soil and water erosion, for these areas have more frequent human activities” meanings not clear, please rephrase it.
  5. Figure 2. Distribution of domestic and foreign research literature. Three categories of literature is mentioned please explain them somewhere which make it easy to understand the figure for the readers.
  6. Expend the conclusion and add few sentences of success stories of problem solution from reviews

Good Luck!   

Author Response

(The authors gave the same response as above.)

Round 2

Reviewer 1 Report

See the attached Comments.

Author Response

(The authors gave the same response as above.)

Reviewer 2 Report

added modifications improve clairly the manuscript 

Author Response

Thank the reviewers for their valuable suggestions. 

Best wishes to you!

Reviewer 3 Report

The MS is properly revised and is now looking fine to me.

Author Response

(The authors gave the same response as above.)
